# Upright and Crawling Locomotion and Its Transition for a Wheel-Legged Robot

**DOI:** 10.3390/mi13081252

**Published:** 2022-08-04

**Authors:** Xuejian Qiu, Zhangguo Yu, Libo Meng, Xuechao Chen, Lingxuan Zhao, Gao Huang, Fei Meng

**Affiliations:** 1School of Mechatronical Engineering, Beijing Institute of Technology, Beijing 100081, China; 2Key Laboratory of Biomimetic Robots and Systems, Ministry of Education, Beijing 100081, China; 3Faculty of Information Technology, Beijing University of Technology, Beijing 100124, China

**Keywords:** wheel-legged robot, two locomotion modes, mode transition, quadratic programming

## Abstract

To face the challenge of adapting to complex terrains and environments, we develop a novel wheel-legged robot that can switch motion modes to adapt to different environments. The robot can perform efficient and stable upright balanced locomotion on flat roads and flexible crawling in low and narrow passages. For passing through low and narrow passages, we propose a crawling motion control strategy and methods for transitioning between locomotion modes of wheel-legged robots. In practical applications, the smooth transition between the two motion modes is challenging. By optimizing the gravity work of the body, the optimal trajectory of the center of mass (CoM) for the transition from standing to crawling is obtained. By constructing and solving an optimization problem regarding the posture and motion trajectories of the underactuated model, the robot achieves a smooth transition from crawling to standing. In experiments, the wheel-legged robot successfully transitioned between the crawling mode and the upright balanced moving mode and flexibly passed a low and narrow passage. Consequently, the effectiveness of the control strategies and algorithms proposed in this paper are verified by experiments.

## 1. Introduction

As a masterpiece of species evolution, the physiological structure of the human body provides a good structural basis for leg movement. A humanoid robot can walk on two legs while freeing hands to complete complex two-arm collaborative tasks [1,2,3,4,5,6,7]. Human-sized robots are easier to adapt to the daily environment that is designed and decorated for humans. Considering the movement capabilities of the legs and arms and the adaptability to the human environment, we believe that a humanoid structure is the best morphological structure choice for robots. However, humanoid robots are less efficient at moving on flat roads than wheeled robots. Therefore, we decided to develop a novel wheel-legged robot, which has the basic body structure and freedom of movement of a humanoid robot and replaces the feet with actuated wheels. Wheel-legged robots combine the advantages of wheeled robots and legged robots [8,9,10], so they can travel efficiently on flat ground with the help of wheels and overcome uneven terrains and obstacles with the help of legs [11,12,13,14,15,16].

In recent years, improving environmental adaptability has become the focus of research in the field of wheel-legged robots [17,18,19,20,21]. The wheel-legged robot is an underactuated system and its stability is greatly affected by the terrain and environment, so it is necessary to have strong environmental adaptability. Handle [22], developed by Boston Dynamics, is a milestone in the development of wheel-legged robots. Handle can pass through slopes, stairs, and jump over obstacles in upright balanced moving mode. Unfortunately, few technical details have been undisguised. Klemm et al. presented the wheel-legged robot Ascento that applied an LQR-assisted whole-body control strategy to stabilize the robot system [23]. Ascento can jump up stairs by way of the mixed motion of legs and wheels. H. Zhou et al. proposed a hydraulic wheel-legged robot WLR [24,25,26] that integrated leg structures and hydraulic oil tubing to improve the movement ability and environmental adaptability of the robot. BIT developed a novel motor-driven wheel-legged robot BHR-W [27] that can travel outdoors on grass and rough roads with the help of a layered controller of legs and wheels. It is worth noting that the wheel-legged robots presented in the above studies have only one mode of motion that allows the robot to perform upright balanced moving in an open environment. However, in practical application scenarios, in addition to the open environment, low and narrow passages are common and important application environments, such as underground pipeline inspection and cave exploration. Therefore, the environmental adaptability of the wheel-legged robot with only one motion mode is limited.

In fact, if the robot can switch between different modes of locomotion according to the different environments, the environmental adaptability of the wheel-legged robot will be greatly improved, especially in unstructured environments coexisting with people. In the DARPA robotics challenge, the Hubo robot [28,29] was equipped with wheels on its knees and feet and it successfully completed the task with the help of the flexible transition between walking mode and wheeled mode in complex environments. However, its transition control strategy is full-drive and quasi-static, which is not suitable for an underactuated wheel-legged robot. Liu et al. [30] developed a wheel-legged robot *SR600-II* that has the structure of the foot and can perform flexible transitions between wheels and feet. In another case, from the released video, the ANYmal robot can achieve the transition from wheel-quadruped mode to wheel-biped mode by using the Multi-AMP algorithm to enhance the deep learning framework [31,32]. Unfortunately, a long time is required to generate motion priors with limited computing power, which is in most cases not available for specific tasks. In general, the mode transition of wheel-legged robots mainly focuses on the walking mode and the wheeled mode, while little research on the crawling mode adapted to low and narrow environments has been published.

We believe that the ability of a wheel-legged robot to work in low and narrow environments is an important indicator to measure its environmental adaptability. Therefore, in this paper we develop a crawling motion mode of the wheel-legged robot for low and narrow environments and propose control strategies for the transition between the crawling mode and upright balanced moving mode. In particular, the standing recovery control of the underactuated wheel-legged robot is a challenging technical task, that enables the robot to transition from crawling to standing.

The main contributions of this paper are as follows:We develop a new wheel-legged robot with two locomotion modes and the ability to transition between them and build simplified models for different motion modes of the robot;We propose a crawling control strategy that can allow the robot to stably pass through low and narrow passages with different heights and different curve radii;We propose locomotion mode transition control methods that enable the robot to transition between the crawling mode and the upright balanced moving mode by respectively optimizing the gravity work of the body and solving a quadratic programming (QP) problem about the posture and motion trajectories of the robot.

The remainder of the paper is organized as follows: Section 2 introduces the mechanical design and hardware of the robot system. Section 3 introduces the modeling and control algorithms of the upright balanced moving mode and crawling mode of the robot. The control methods of transition between locomotion modes are presented in Section 4. The proposed approaches are verified by experiments on the BHR-W robot in Section 5. Finally, we draw conclusions and discuss future work in Section 6.

## 2. BHR-W Robot System

### 2.1. Mechanical Design

The purpose of mechanical design is to develop a wheel-legged robot with human-like proportions, range of movement and kinematic structure. Therefore, the kinematic scheme for the robot is planned with 14 actuated DoFs based on human-like motion capabilities and degrees of freedom configuration, as shown in Figure 1. Each leg has four DoFs including hip pitch and roll, knee pitch and an actuated wheel. Each arm has three DoFs and a passive wheel at the wrist. The robot is 1.61 m tall and the weight of the whole system is 55 kg.

We analyse the range of movement of the robot’s arms and legs with the help of kinematic analysis, as shown in Figure 2. The results show that the robot’s size and degrees of freedom configuration enable the robot to have a large range of movement, which meet the needs of human-like motion capabilities. The analysis results can assist researchers in motion planning and control within a reasonable range of movement, as shown in Table 1.

The leg structure and freedom of movement configuration are designed in series as shown in Figure 3a. To reduce the inertia of the leg and improve the response speed, the leg motors are intensively placed in the hip and control the flexion, extension and swing movements of the leg. We integrate the motor and reducer into the leg structure, which increases the range of leg movement and reduces weight. The design scheme greatly simplifies the leg structure and saves space that is originally prepared for the drive unit. The torque that drives the motion of the knee joint comes from the motor torque transmitted by the linkages. The length ratio of the torque transmission linkages is obtained by numerical optimization, which can guarantee meeting the torque requirements of the knee joint in the available range of movement.

Each arm is divided into two parts: the upper arm and the forearm, as shown in Figure 3b. The upper arm can perform pitch and roll movements. At the elbow joint, the motor drives the ball screw mechanism to perform linear motion to drive the flexion and extension of the forearm. The elbow joint can achieve a range of movement from 30 degrees to 150 degrees. During the movement of the elbow joint, the angle δ between the screw push rod and the forearm is constantly changing, which causes the torque acting on the elbow joint is also changing. Therefore, we perform a kinematic analysis on the relationship between elbow joint angle and torque at rated output torque condition, as shown in Figure 4. It can be seen from the curve that in the range of movement from 50 degrees to 110 degrees, the torque value of the elbow joint is larger than 45, which can guide the motion planning of the elbow joint to make it move within a suitable torque range.

Actuated wheels at the end of each leg are directly driven by the hub motor, and passive wheels are installed at the wrist. The actuated wheels provide power and stability for the high-speed balanced moving, while the passive wheels assist the robot in changing direction when crawling on all limbs. Additionally, the framework of the robot is mainly fabricated from aluminium alloy, titanium alloy and carbon fiber to achieve high stiffness and maintain lightweight.

### 2.2. Hardware System

The hardware system includes drive units, sensor units, communication systems and a computer. Considering the mass of the robot and the impact from uneven ground, the drive units need to provide a large torque. Therefore, we customize the frameless torque motors and planetary reducers according to the size and torque requirements. All motors communicate with the onboard computer via EtherCAT. The onboard computer is an Intel NUC 8 and the processor is an i7-8559U. The inertial measurement unit(IMU) is installed in the center of the torso and its feedback information plays an important role in the state estimation of the robot. Furthermore, in order to help us evaluate the contact state between the actuated wheels and the ground, a six-dimensional force sensor is integrated into the ankle.

## 3. Modeling and Control of Two Locomotion Modes

The wheel-legged robot BHR-W has two motion modes: the upright balanced moving mode on two wheels and the crawling mode on four limbs. Next, we construct the kinematic and rigid body dynamic models of the above locomotion modes.

### 3.1. Coordinates

To clearly describe the locomotion of the robot, we define the coordinate systems and variables as shown in Figure 5. The forward direction of the robot is the positive direction of the *x*-axis. The vertical upward direction is the positive direction of the *z*-axis. The *y*-axis is parallel to the wheel axis. θ denotes the tilt angle of the CoM relative to the *z*-axis. γ denotes the angle between the forward direction of the robot and the *y*-axis at the inertial coordinate system $\sum_{0}$.

### 3.2. Modeling and Control of the Upright Balanced Moving Mode

#### 3.2.1. Upright Balanced Moving Model

The upright balanced locomotion of the wheel-legged robot is a complex motion formed by the coupling of wheel motion and body motion. Therefore, it is necessary to build a layered control model for the robot [27].

The main purpose of wheel motion control is to achieve upright movement on the basis of maintaining body balance and stability. A two-wheeled inverted pendulum model [33,34,35] is built for the upright balanced moving control as shown in Figure 5a. The torso, arms and legs are simplified as a rigid body on wheels. The height of the CoM and the moment of inertia of the rigid body are provided by the body motion model.

Combining defined variables and the physical parameters of the model, the kinetic and potential energies of the simplified model, *T* and *V*, can be derived.
(1)$$\left\{\begin{array}{c} T\left(\varphi ,\dot{\varphi }\right)=T_{wheel}\left(\dot{\varphi }\right)+T_{body}\left(\varphi ,\dot{\varphi }\right)\\ V\left(\varphi \right)=V_{wheel}+V_{body}\left(\varphi \right), \end{array}\right.$$
where φ is the model generalized coordinate containing θ, $\phi_{L}$ and $\phi_{R}$.

The Lagrangian energy function given by:(2)$$L\left(\varphi ,\dot{\varphi }\right)=T\left(\varphi ,\dot{\varphi }\right)-V\left(\varphi \right).$$

The dynamic equations of the model are obtained by using the Euler–Lagrange equations:(3)$$\frac{d}{dt}\left(\frac{\partial L}{\partial \dot{\varphi }}\right)-\left(\frac{\partial L}{\partial \varphi }\right)=M,$$
where *t* is the continuous time variable and *M* is the generalized torque.

Finally, the dynamic equation of the two-wheeled inverted pendulum model can be expressed as:(4)$$\begin{array}{c} D_{w}\left(\varphi \right)\ddot{\varphi }+C_{w}\left(\varphi ,\dot{\varphi }\right)\dot{\varphi }+G_{w}\left(\varphi \right)=\tau_{w}\\ \tau_{w}={\left[\left(M_{f}-M_{L}-M_{R}\right)\;\;\;\;\left(M_{f}-M_{L}\right)\;\;\;\;\left(M_{f}-M_{R}\right)\right]}^{T}, \end{array}$$
where $\varphi ={\left[\theta \;\;\;\phi_{L}\;\;\;\phi_{R}\right]}^{T}$, $D_{w}\in R^{n\times n}$ denotes the mass matrix, $C_{w}\in R^{n}$ denotes the vector of coriolis and centrifugal terms, and $G_{w}\in R^{n}$ is the vector of gravity term. $M_{R}$ and $M_{L}$ respectively indicate the output torque of the two wheel motors. $M_{f}$ is the friction torque on wheels.

Body motion can change the height of the CoM and the moment of inertia of the two-wheeled inverted pendulum model. Therefore, we simplify the symmetrical body to a 3-link model as shown in Figure 5b, and the physical parameters are shown in Table 2.

The position of the CoM and the moment of inertia of the model can be expressed as:(5)$$P_{CoM}\left(\beta \right)=\frac{\sum_{i}^{}m_{i}\cdot P_{CoM}^{i}\left(\beta \right)}{M}$$
(6)$$I_{\left\{\theta ,\gamma \right\}}=\sum_{i}^{}m_{i}r_{\left\{iy,iz\right\}}^{2},$$
where $P_{CoM}\left(\beta \right)$ is the position of the CoM, $\beta ={\left[\beta_{1}\;\;\;\beta_{2}\;\;\;\beta_{3}\;\;\;\beta_{4}\right]}^{T}$ is the generalized coordinate of the 3-link model, and $r_{\left\{iy,iz\right\}}$ are the distance from the CoM of each part to the *y*-axis or the *z*-axis in coordinate system $\sum_{w}$.

#### 3.2.2. Upright Balanced Moving Control

The wheel-legged robot is an underactuated system. A reliable balance controller based on the linear quadratic regulator (LQR) and the body motion controller is developed and applied to the upright balanced moving control of the wheel-legged robot, as shown in Figure 6. Li et al. [36] proved that an LQR controller can provide highly reliable and robust balance control for a two-wheeled inverted pendulum system.

The state equation of the robot system is obtained by linearizing the dynamic equations of the simplified model in Equation (Equation 4).
(7)$$\left\{\begin{array}{c} \dot{x}=Ax+Bu\\ y=Cx, \end{array}\right.$$
where $x={\left[\theta \;\phi_{L}\;\phi_{R}\;\dot{\theta }\;\dot{\phi_{L}}\;\dot{\phi_{R}}\right]}^{T}$, $u={\left[M_{L}\;M_{R}\right]}^{T}$, $A\in R^{6\times 6}$ is the system matrix, $B\in R^{6\times 2}$ is the input matrix.

The problem solved by LQR controller is to obtain the optimal feedback gain matrix *K* so that there is an optimal control variable $u{\left(t\right)}^{*}=-Kx\left(t\right)$ to minimize the quadratic performance index function:(8)$$\min J=\int_{0}^{\infty }\left(x^{T}\left(t\right)Qx\left(t\right)+u^{T}\left(t\right)Ru\left(t\right)\right)dt.$$

By solving the discrete-time algebraic Riccati equation, the feedback gain matrix *K* of the system is obtained:(9)$$A^{T}P+PA-PBR^{-1}B^{T}P+Q=0$$
(10)$$K=R^{-1}B^{T}P,$$
where *Q* and *R* are the weight matrices and *P* is the solution of the Riccati Equation (Equation 9).

Finally, the optimal input variable that can control the stability of upright balanced locomotion is obtained:(11)$$u=-K\hat{x},$$
where the optimal input variable $u={\left[M_{L}\;M_{R}\right]}^{T}$ is the output torque of the two wheel motors. The state variable of the system is $\hat{x}$ supplied by the state estimator.

It is worth emphasizing that *Q* and *R* are constructed based on the importance of the state variables and the controlled variables. The larger the value on the diagonal of the *Q* matrix, the faster the system responds to the corresponding state variable. *R* is the weight matrix of the controlled variables, and the larger the value on its diagonal is, the stronger the constraint on the controlled variable. We gradually adjusted the weight value in experiments until the robot system showed strong stability and robustness.

However, in the actual upright balanced movement on two wheels, the height of the robot’s CoM *h* often changes due to body motions, such as squatting and standing up. Therefore, we need to continuously adjust the parameters of the LQR balance controller to ensure that the robot is stable and reliable in the upright balanced moving mode.

The controller calculates *h* the height of the CoM, $I_{\gamma }$ the moment of inertia of body around the *z*-axis and $I_{\theta }$ the moment of inertia of body around the wheel axis every five periods with the help of the dynamics library.
(12)$$\begin{array}{cc} \; & h=\sqrt{{\left(P_{CoM}^{x}\left(\beta \right)\right)}^{2}+{\left(P_{CoM}^{z}\left(\beta \right)\right)}^{2}} \end{array}$$
(13)$$\begin{array}{cc} \; & I_{\theta }=I_{CoM}^{y}+Mh^{2}\; \end{array}$$
(14)$$\begin{array}{cc} \; & I_{\gamma }=I_{CoM}^{z}+M{\left(P_{CoM}^{x}\left(\beta \right)\right)}^{2},\; \end{array}$$
where the value of $I_{CoM}^{\left\{x,y,z\right\}}$ are the moment of inertia of body around the CoM in {x,y,z} direction given by the dynamics library.

They are fed into the LQR balance controller to obtain the real-time feedback gain matrix $\tilde{K}$. Finally, the balance controller uses the $\tilde{K}$ matrix and $\hat{x}$ fed back by the state observer to obtain the real-time optimal input variable $\tilde{u}$ for the upright balanced moving mode.
(15)$$\tilde{u}=-\tilde{K}\hat{x}.$$

### 3.3. Modeling and Control of the Crawling Mode

#### 3.3.1. Crawling Motion Model

The main goal of crawling on four limbs is to pass through low and narrow passages. It is necessary for the crawling motion model to have the ability to change the height of the torso from the ground and to achieve flexible turning. The crawling motion model with the help of the hybrid locomotion of four limbs is shown in Figure 7. The dimensions of each body part are shown in Table 3.

The height of the torso from the ground $H_{torso}$ can be derived as:(16)$$H_{torso}=L_{1}^{arm}\sin\left(\alpha_{2}\right)+L_{2}^{arm}\sin\left(\alpha_{2}+\alpha_{3}\right)+L_{aw}=L_{1}^{leg}\sin\left(\beta_{2}\right)+L_{2}^{leg}\sin\left(\beta_{2}+\beta_{3}\right)+R_{w}.$$

The turning radius $R_{turn}$ can be expressed as:(17)$$\begin{array}{cc} \; & R_{turn}=\frac{L_{T}}{\sin\beta_{1}}\; \end{array}$$(18)$$\begin{array}{cc} \; & L_{T}=L_{torso}+L_{arm}+L_{leg}\cos\beta_{1}\; \end{array}$$(19)$$\begin{array}{cc} \; & L_{arm}=-L_{1}^{arm}\cos\alpha_{2}-L_{2}^{arm}\cos\left(\alpha_{2}+\alpha_{3}\right) \end{array}$$(20)$$\begin{array}{cc} \; & L_{leg}=L_{1}^{leg}\cos\beta_{2}+L_{2}^{leg}\cos\left(\beta_{2}+\beta_{3}\right).\; \end{array}$$

#### 3.3.2. Crawling Motion Control

When crawling as shown in Figure 7, wheels equipped at the ends of four limbs of the robot touch the ground. The state of touching the ground at four points is reliable and stable. Therefore, in this mode, we do not perform additional control for stability and only control the movement of the robot such as forward, backward, turning, etc.

In the crawling motion mode, the drive wheel motors are in the speed control mode. When driving on a straight road, the controller will drive the wheel motors to reach the reference velocity *v*. When passing through a curve, the operator can control and drive the roll angle of the hip joint to make the robot turn. During turning as shown in Figure 7b, the angle between the upper body and the legs is less than 180 degrees and the CoM is in the stable region. If the radius of the curve is less than the minimum turning radius of the robot, the differential driving of the wheels can be activated to assist the robot in successfully passing through the curve successfully.

In addition, to enable the robot to pass through passages with different heights, the distance $H_{torso}$ between the torso and the ground can also be controlled by changing the each joint angle of the four limbs. The passive wheels are installed at the wrist, so it is necessary to ensure that the forearm and calf are parallel to the ground as shown in Figure 7a: (21)$$\begin{array}{c} \alpha_{3}=\pi -\alpha_{2}\\ \beta_{3}=-\beta_{2}. \end{array}$$

By combining Equations (16) and (21), we can obtain each joint angle of the four limbs, which can help the robot to pass through passages of different heights.

## 4. Modeling and Control of Mode Transition

### 4.1. Modeling and Control of Kneeling

#### 4.1.1. Kneeling Model

The kneeling process is a transition from the upright balanced moving mode to the kneeling state, as shown in Figure 8. The kneeling model of the robot is the same as Figure 5b, in which the upper body and arms are simplified as a rigid body. After kneeling, the angle between the calf and the ground is determined:(22)$$\beta_{4}=\arcsin\left(\frac{R_{w}}{L_{2}^{leg}}\right).$$

The relative position of the CoM of the simplified model can be expressed as:(23)$$P_{CoM}^{\left\{x,z\right\}}\left(\beta \right)=\frac{\sum_{i}^{}m_{i}\cdot P_{CoM}^{\left\{ix,iz\right\}}\left(\beta \right)}{M},$$
where β is the joint angle of the kneeling down model, and *M* is the total mass of the robot. $P_{CoM}^{\left\{x,z\right\}}\left(q\right)$ is the position of the CoM after kneeling down, $P_{CoM}^{i}\left(q\right)$ represent the position of the CoM of each part, and $m_{i}$ represent the mass of each part.

#### 4.1.2. Kneeling Control

The kneeling process of the underactuated wheel-legged robot is similar to a forward falling process as shown in Figure 8. During this process, the controller stops the wheels and causes the robot to lean forward and fall. Therefore, in order to maintain the stability of the system, the mode transition controller needs to ensure that the CoM of the robot is located in the stable region formed by the wheels and the knee after kneeling.
(24)$$p_{min}^{x}\leq P_{CoM}^{x}\left(\beta \right)\leq p_{max}^{x},$$
where $p_{min}^{x}$ and $p_{max}^{x}$ are the boundary coordinates of the stable region.

It is necessary to ensure that the velocity of the CoM in the *x*-axis direction is as small as possible to prevent further falling. In addition, we need to optimize the velocity of the CoM in the *z*-axis direction to reduce the impact force on the knee joint.
(25)$$\begin{array}{cc} \; & \min\left|w_{z}{\dot{P}}_{CoM}^{z}\left(\dot{\beta },\beta \right)+w_{x}{\dot{P}}_{CoM}^{x}\left(\dot{\beta },\beta \right)\right|\;\;\; \end{array}$$
(26)$$\begin{array}{cc} \Rightarrow \; & \min\left|w_{z}\left({\hat{P}}_{CoM}^{z}\left(\hat{\beta }\right)-P_{CoM}^{z}\left(\beta \right)\right)+w_{x}\left({\hat{P}}_{CoM}^{x}\left(\hat{\beta }\right)-P_{CoM}^{x}\left(\beta \right)\right)\right|, \end{array}$$
where $w_{z}$ and $w_{x}$ are the weights of the velocity in the *z*-axis direction and the *x*-axis direction respectively, and ${\hat{P}}_{CoM}^{\left\{x,z\right\}}\left(\hat{\beta }\right)$ is the initial position of the CoM.

We construct an optimization problem about the average velocity of the CoM during the kneeling process, and we obtain the optimal kneeling position and posture that satisfies various demand constraints:(27)$$\begin{array}{cc} \; & \min\;\;\;\;\;\left|w_{z}\left({\hat{P}}_{CoM}^{z}\left(\hat{\beta }\right)-P_{CoM}^{z}\left(\beta \right)\right)+w_{x}\left({\hat{P}}_{CoM}^{x}\left(\hat{\beta }\right)-P_{CoM}^{x}\left(\beta \right)\right)\right|\\ \; & s.t.\;\left\{\begin{array}{c} p_{min}^{x}\leq P_{CoM}^{x}\left(\beta \right)\leq p_{max}^{x}\\ \beta_{min}\leq \beta \leq \beta_{max}\\ \beta_{4}=\arcsin\left(\frac{R_{w}}{L_{2}^{leg}}\right). \end{array}\right. \end{array}$$

### 4.2. Modeling and Control of Standing Recovery

#### 4.2.1. Standing Recovery Model

The standing recovery process is the transition from the crawling mode to the upright balanced moving mode. The most critical motion in this process is from kneeling to standing as shown in Figure 9. The entire kneeling-to-standing motion is similar to swinging up and stabilizing a second-order inverted pendulum. In this process, there is no relative motion between drive wheels and the calf. Wheels only roll on the ground. Therefore, we simplify the robot model to a special second-order inverted pendulum with an underactuated joint that hinges on the ground. The thigh and calf structures are regarded as the first-order link, and the torso and arms are regarded as the second-order link.

It is assumed that the left and right sides of the robot system are completely symmetrical and the mass distribution is uniform. Additionally, using the Euler–Lagrange equations, the dynamic equation of the special inverted pendulum can be derived as:(28)$$D_{r}\left(q\right)\ddot{q}+C_{r}\left(q,\dot{q}\right)\dot{q}+G_{r}\left(q\right)=S_{r}\tau_{r},$$
where $q={\left[q_{1}\;\;\;q_{2}\right]}^{T}$ is the joint angle of the under-actuated link model, $\tau_{r}$ is the torque of all joints, $S_{r}\in R^{n\times n}$ is the control selection matrix.

#### 4.2.2. Standing Recovery Control

The motion planning and control of standing recovery is inspired by the instinctive motions of humans from crawling to standing up. First, a predefined and planned action sequence is triggered from the motion library. During the execution of the action, the trajectory is locally online optimized according to the starting conditions of the balance controller. Therefore, the standing recovery controller is composed of a feed-forward motion library and a trajectory optimization controller. In the following, the standing recovery controller of the robot will be described in further detail.

From Crawling to Kneeling: Inspired by human actions and considering the DoFs of the robot, the humanoid limb motion trajectories are planned and stored in the motion library. The controller makes the joint motors follow the preplanned specific trajectories, so that the robot performs the mode transition to the kneeling state as shown in Figure 10.

From Kneeling to Standing: Based on the simplified underactuated inverted pendulum model and humanoid motion, we divide the transition from kneeling to standing into two stages.

The purpose of the first-stage control, as shown in Figure 9a,b, is to move the CoM of the robot back to above the wheel axis vertically and to make the knee joint leave the ground. Therefore, we need to plan the motions and control the upper body of the robot to lean back according to the desired position of the CoM. Then, we use the kinematic method to map the CoM trajectory to all joints.
(29)$${}_{}^{b}{\dot{P}}_{CoM}={}_{}^{b}J_{CoM}\dot{q},$$
where ${}_{}^{b}{\dot{P}}_{CoM}$ is the velocity of the CoM in the coordinate system $\sum_{w}$, ${}_{}^{b}J_{CoM}$ is the Jacobian matrix of the CoM, and $\dot{q}$ is the joint angular velocity.

The upper body of the robot performs a backward leaning motion to move the CoM of the robot backward. Due to the existence of underactuated joint, when the CoM reaches above the wheel axis, the robot will lean back around the underactuated joint with a certain angular velocity.

Since the robot still has a certain angular velocity rotating around the underactuated joint, the system state is unstable, so the second-stage control takes effect. The purpose of the second-stage control is to adjust the robot posture to meet the starting conditions of the balance controller. The mechanical median position is a special posture where the robot can self-balance without control and the CoM is vertically above the wheel axis. The two-wheeled balance controller needs to be triggered when the robot posture is close to the mechanical median position, otherwise the robot may oscillate due to overshoot. In addition, if the robot still has a large angular velocity rotating around the wheel axis when the balance controller is activated, it will cause a huge challenge to the stability of the robot. Therefore, the goal of the standing recovery controller is to ensure that robot reaches the mechanical median position and has no angular velocity around the wheel axis. Although the CoM of the robot is above the wheel axis vertically at the end of the first-stage control, the robot has a relatively large angular velocity rotating around the wheel axis. Clearly, this does not satisfy the starting conditions of the balance controller. Therefore, the state estimator needs to feed back the angular velocity information of the robot rotating around the wheel axis as the state parameters for the second-stage control.

Because the robot has angular velocity of backward rotation, the CoM will move back away from the mechanical median position. Each control cycle predicts whether the CoM of the robot will deviate from the mechanical median position in the next cycle based on the position of the CoM and the angular velocity information. If it is predicted that there will be a large deviation between the position of the CoM and the mechanical median position, the controller will plan the trajectory online to make the CoM return back. If the deviation is small enough to be within a certain threshold range, it is considered that the starting condition of the balance controller is satisfied.

Predict the angle of the underactuated joint in the next cycle and plan the motion trajectory of the actuated joint based on it to make the robot reach back to the mechanical median position. Since the model is underactuated, the robot cannot accurately perform the planned trajectory, so we need to use the optimization method to obtain the optimal trajectory and torque of the actuated joints combined with the online planning trajectory and the dynamic Equation (Equation 28).

By constructing and solving a QP problem [37] about the posture and motion trajectories of the underactuated model, we obtain the optimal joint trajectory and torque for the robot to transition from crawling to standing.
(30)$$\begin{array}{cc} \; & \min\;\;\;\;\;\frac{1}{2}{∥AX-b∥}^{2}=\frac{1}{2}X^{T}\left(A^{T}A\right)X+{\left(-A^{T}b\right)}^{T}X+\frac{1}{2}b^{T}b\\ \; & s.t.\;\left\{\begin{array}{c} \left[C\left(q,\dot{q}\right)\;\;\;\;D\left(q\right)\;\;\;-S_{1}\right]X+G\left(q\right)=0\\ {\dot{q}}_{min}\leq S_{2}X\leq {\dot{q}}_{max}\\ \tau_{min}\leq S_{3}X\leq \tau_{max}\\ q_{min}\leq q\leq q_{max}. \end{array}\right. \end{array}$$

The optimization variable is $X={\left[\dot{q}\;\;\ddot{q}\;\;\tau \right]}^{T}$. $S_{i}$ is the selection matrix and the equation constraint is the dynamic equation of the underactuated model, which ensures that the motion of the underactuated joint follows the laws of physics. Inequality constraints provide the range of joint angles and torque limits.

In practical applications, we use Eigen-QuadProg [38] to solve the QP problem online. When the predicted deviation of the centroid position from the mechanical median position is small enough to be within a certain threshold range, the robot will end the trajectory optimization control and start the balance controller.

The purpose of the cost function is to ensure that the motion of the robot is close to the desired position and velocity, as follows:(31)$$\begin{array}{cc} \; & A=diag(w_{{\dot{q}}_{1}},w_{{\dot{q}}_{2}},w_{{\ddot{q}}_{1}},w_{{\ddot{q}}_{2}},w_{\tau_{1}},w_{\tau_{2}})\\ \; & b=\left[\begin{array}{c} O_{2\times 1}\\ {\ddot{q}}_{1}^{ref}\\ {\ddot{q}}_{2}^{ref}\\ O_{2\times 1} \end{array}\right]=\left[\begin{array}{c} O_{2\times 1}\\ k_{p1}\left(q_{1}-q_{1}^{ref}\right)+k_{d1}\left({\dot{q}}_{1}-{\dot{q}}_{1}^{ref}\right)\\ k_{p2}\left(q_{2}-q_{2}^{ref}\right)+k_{d2}\left({\dot{q}}_{2}-k{\dot{q}}_{1}\right),\\ O_{2\times 1} \end{array}\right] \end{array}$$
where $w_{i}$ represents the weight of the specific desired behavior. The penalty for deviating from the desired state is reflected in the acceleration term. $q^{ref}$ and ${\dot{q}}^{ref}$ from motion trajectories are planned by a higher level controller. In addition, *k* represents the specific angular velocity relationship between the drive rod and the under-actuated rod, whose purpose is to restrain the angular velocity of the system as small as possible.

When the robot reaches the mechanical median position, the CoM is vertically above the wheel axis. Therefore, at this moment, the angular momentum of the under-actuated system is conserved.
(32)$$\sum_{i}^{}I_{i}\omega_{i}=I\omega ,$$
where $I_{i}$ is the moment of inertia of each part around the under-actuated joint, $\omega_{i}$ is the angular velocity of each part, *I* is the moment of inertia of the whole system around the under-actuated joint, and ω is the angular velocity of the whole system.

ω as small as possible is one of the starting conditions of the balance controller. So the angular velocity relationship *k* can be expressed as:(33)$$k=\frac{I_{1}}{I_{2}}.$$

## 5. Experiments

In this section, we introduce the traveling experiments of BHR-W in two locomotion modes and the mode transition experiments between them. The control rate of the robot is 1 kHz. The motion commands for the robot, such as forward, backward and turn, and the mode transition commands are all controlled by the operator remotely.

### 5.1. Upright Balanced Moving Experiment

We control the BHR-W to carry out the upright balanced moving experiment on an outdoor road. Figure 11 presents snapshots of the upright balanced moving experiment. The robot successfully completes the motion commands input by the operator and maintains the balance and stability of the robot system during the moving process with the help of the balance controller. Motion commands are input remotely by the operator, and the balance controller runs autonomously on the onboard computer.

The data curves are shown in Figure 12, which shows that the operator controls the robot to accelerate forward and then decelerate, and turn in the balanced standing state. The value of θ shows that the body leans forward when accelerating and leans backward when decelerating. The experiment shows that the balance controller is stable and reliable.

### 5.2. Crawling Experiment

To demonstrate that the robot can pass through low and narrow passages and to show the reliability of the proposed motion control strategy for crawling, we perform a crawling experiment. As shown in Figure 13, with the assistance of the operator, the BHR-W passes through a channel with a size of 30 cm × 80 cm × 60 cm and makes a large radius turn.

### 5.3. Kneeling Experiment

The robot completes the transition from the upright balanced moving mode to the kneeling posture as shown in Figure 14. In the experiment, we find that the tilting and falling direction of the robot can not be accurately controlled after the wheel motor stops. Therefore, we control the wheel motor to rotate backward by a small angle, so that the robot has a tendency to fall forward. Figure 15 shows that the position of the CoM is in the stable region ($P_{CoM}^{x}$ > 0.078 m) after kneeling on the ground. At the moment of hitting the ground, $v_{x}$ and $v_{z}$ are within the acceptable range. The oscillation of $P_{CoM}^{x}$ and $P_{CoM}^{z}$ is caused by the elastic deformation of the tyre. At the beginning of the experiment, the sudden change in $v_{x}$ is caused by the backward rotation of the wheel motor. In summary, the controller stably completes the mode transition task from the upright balanced moving mode to the kneeling state and reduces the impact force on the knee joint.

### 5.4. Standing Recovery Experiment

First, the prototype performs the preplanned motion trajectories to complete the transition from crawling on four limbs to kneeling as shown in Figure 16.

Then, with the help of the mode transition controller, the BHR-W transitions from kneeling to the upright balanced moving mode as shown in Figure 17. During the process shown from Figure 17a–d, the controller causes the upper body to lean back and leaves the knee joint off the ground. During the process shown from Figure 17e,f, with the assistance of the QP online controller, the posture of the underactuated robot reaches the mechanical median position and the balance controller is activated.

As seen in Figure 18, the CoM moves rapidly backwards and eventually stabilizes at approximately 0. $P_{CoM}^{x}=0$ means that the CoM is vertically above the wheel axis, which proves that the robot posture reaches the mechanical median position. The velocity curve shows that $v_{x}$ is close to 0 when arriving at the mechanical median position. The above data analysis shows that the state of the robot satisfies the starting conditions of the balance controller, so the position of the CoM does not have a large oscillation. In general, the experiment demonstrates the effectiveness of the standing recovery controller.

Finally, the experiment of the robot standing up, which changes the height of the CoM, is shown in Figure 19. Figure 20 records the position and velocity data of the CoM relative to the coordinate system $\sum_{w}^{}$ during the standing process. The height of the CoM steadily increases from 0.48 m to 0.72 m. $P_{CoM}^{x}$ fluctuates around 0 mm and finally stabilizes at approximately 2 mm, and $v_{x}$ does not exceed ±0.03 m/s. In summary, the data show that the CoM of the robot has no obvious overshoot during standing up.

The balance controller combines the upper body motion controller and the LQR controller to update the feedback gain matrix *K* every control cycle to achieve stable balance control in response to the robot’s centroid height variation in standing up experiment, which proves that the balance controller has a certain robustness.

## 6. Conclusions

This paper presents a novel wheel-legged robot BHR-W and control algorithms for two locomotion modes and mode transitions. Through the cooperation of the LQR balance controller and the body motion controller, BHR-W can achieve stable and efficient upright balanced locomotion on two wheels. The crawling control algorithm allows the robot to flexibly pass through low and narrow passages with different heights and different curve radii. In addition, the mode transition of the robot can be realized by the optimal control of the underactuated model. In the experiments, the robot completes the locomotion of the upright balanced moving mode and the crawling mode and the mode transition between them. The effectiveness and robustness of the proposed control algorithms have been validated by the experimental results. We believe that it is necessary for the robot to switch motion modes when facing different terrains and environments. Therefore, the control methods of the wheel-legged robot proposed in this paper have strong potential for practical application.

In the future, we will equip the robot with a vision system so that it can autonomously recognize different environments and switch motion modes to adapt to the environment.

## Figures and Tables

**Figure 1 micromachines-13-01252-f001:**
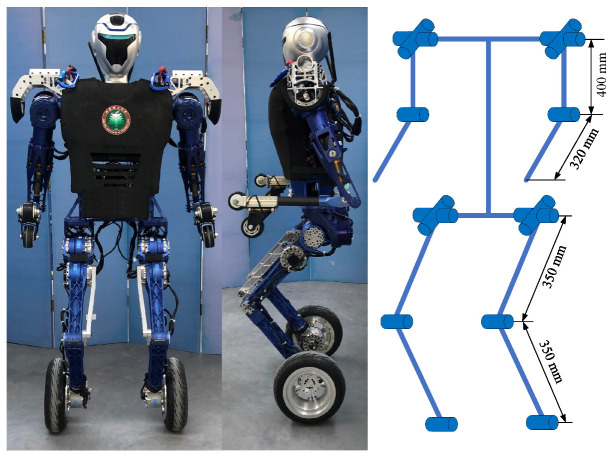
The wheel-legged robot BHR-W.

**Figure 2 micromachines-13-01252-f002:**
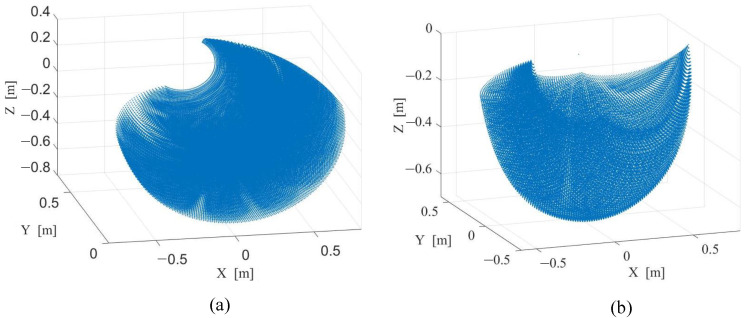
Cloud maps of the range of movement of the robot’s limbs. (**a**) The cloud map of arms; (**b**) The cloud map of legs.

**Figure 3 micromachines-13-01252-f003:**
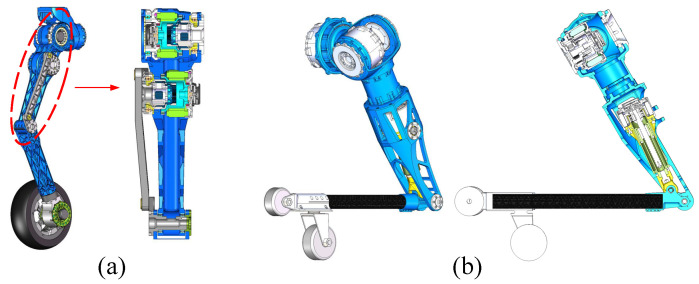
Three-dimensional (3D) structural model of the robot limbs. (**a**) Integrated structure of the robot leg and motors; (**b**) Structure of the robot arm.

**Figure 4 micromachines-13-01252-f004:**
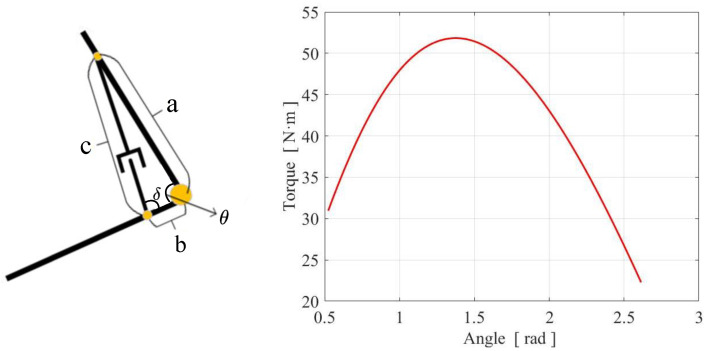
Kinematic analysis of the relationship between elbow joint angle and torque.

**Figure 5 micromachines-13-01252-f005:**
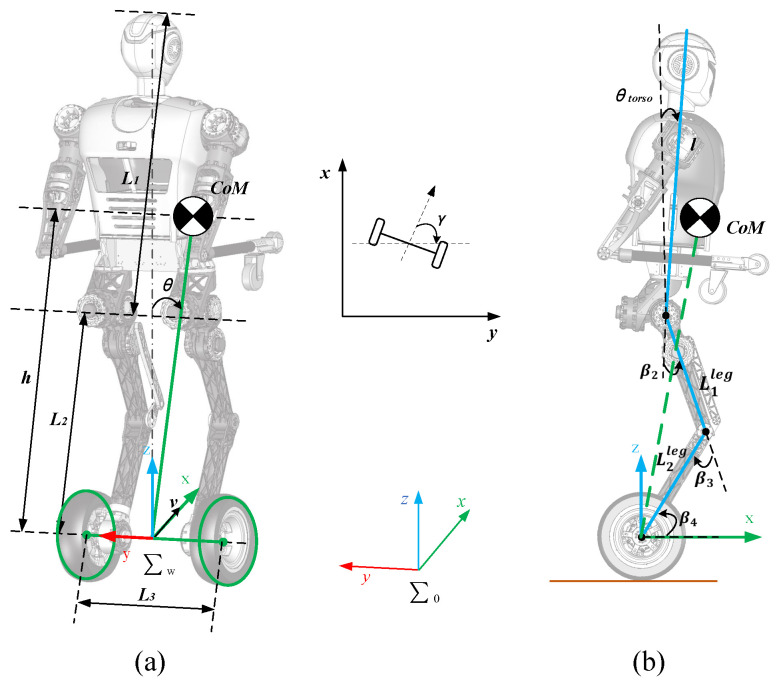
Generalized coordinates of the wheel-legged robot model: (**a**) The inverted pendulum model on wheels; (**b**) The simplified model of body locomotion.

**Figure 6 micromachines-13-01252-f006:**
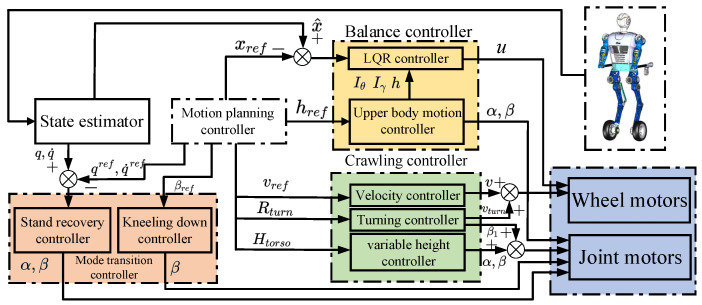
Overview of the controller architecture including two-wheeled balance controller, crawling on four limbs controller and mode transition controller. $\alpha ={\left[\alpha_{1}\;\;\alpha_{2}\;\;\alpha_{3}\right]}^{T}$ is the angle of each joint of the arm.

**Figure 7 micromachines-13-01252-f007:**
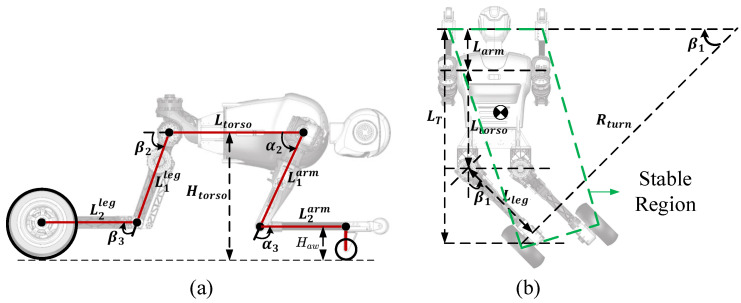
The model of limbs crawling motion: (**a**) The variable height crawling model; (**b**) The crawling turning model.

**Figure 8 micromachines-13-01252-f008:**
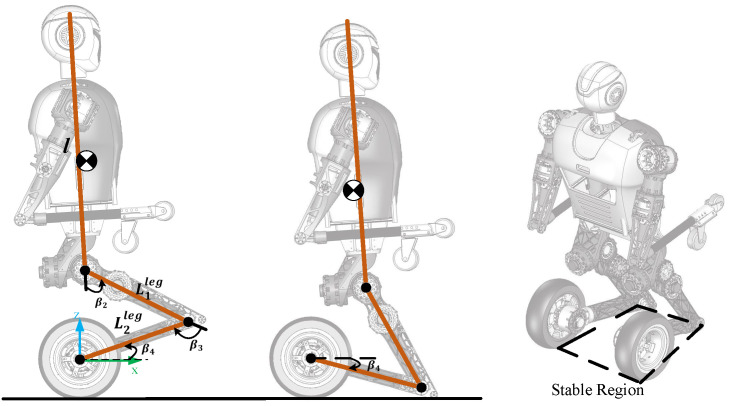
The model of kneeling down.

**Figure 9 micromachines-13-01252-f009:**
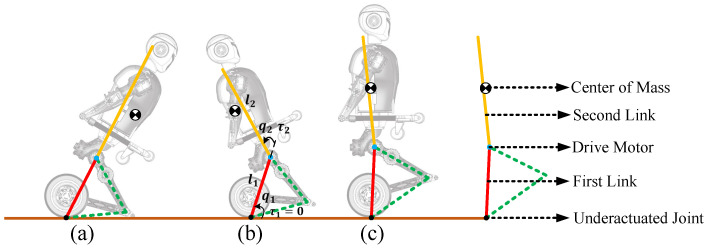
An under-actuated second-order inverted pendulum model: (**a**) The kneeling posture; (**b**) The leaning back posture; (**c**) The squatting posture.

**Figure 10 micromachines-13-01252-f010:**
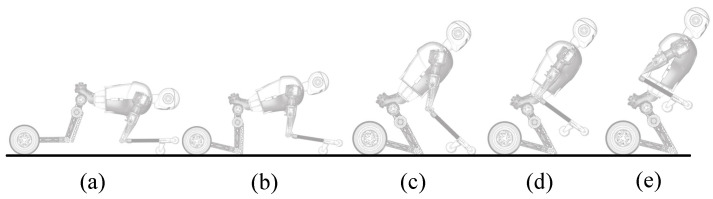
The transition process from crawling to kneeling (**a**–**e**).

**Figure 11 micromachines-13-01252-f011:**
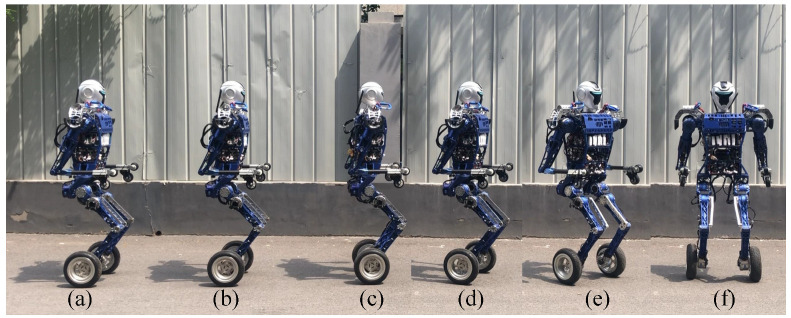
Upright balanced moving experiment on the outdoor road: (**a**–**c**) Forward experiment; (**d**–**f**) Turning experiment.

**Figure 12 micromachines-13-01252-f012:**
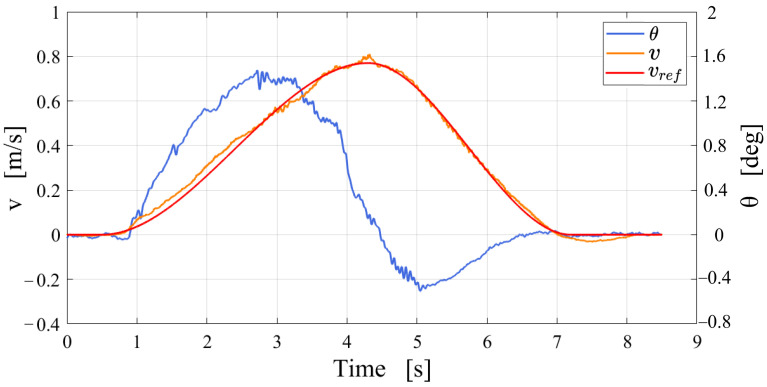
Date graph of the upright balanced moving experiment on the outdoor road: *v* is the forward velocity of the robot; θ is the angle between the CoM and the *z*-axis.

**Figure 13 micromachines-13-01252-f013:**
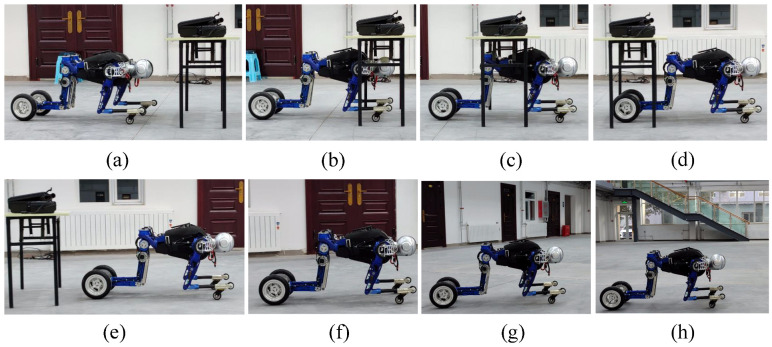
The robot crosses a channel with a size of 30 cm × 80 cm × 60 cm and performs a turn (**a**–**h**).

**Figure 14 micromachines-13-01252-f014:**
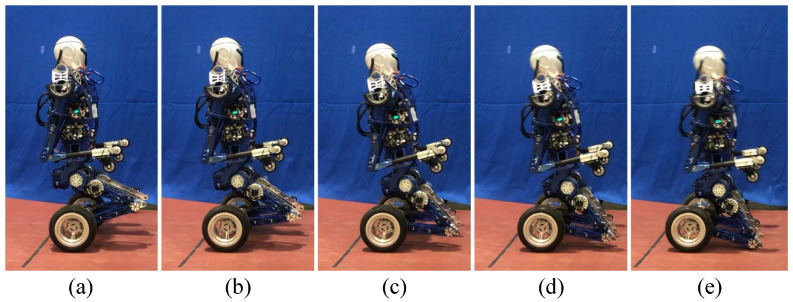
The transition experiment of the robot from the upright balanced moving mode to kneeling on the ground: (**a**) Balanced standing; (**b**–**e**) From balanced standing to kneeling.

**Figure 15 micromachines-13-01252-f015:**
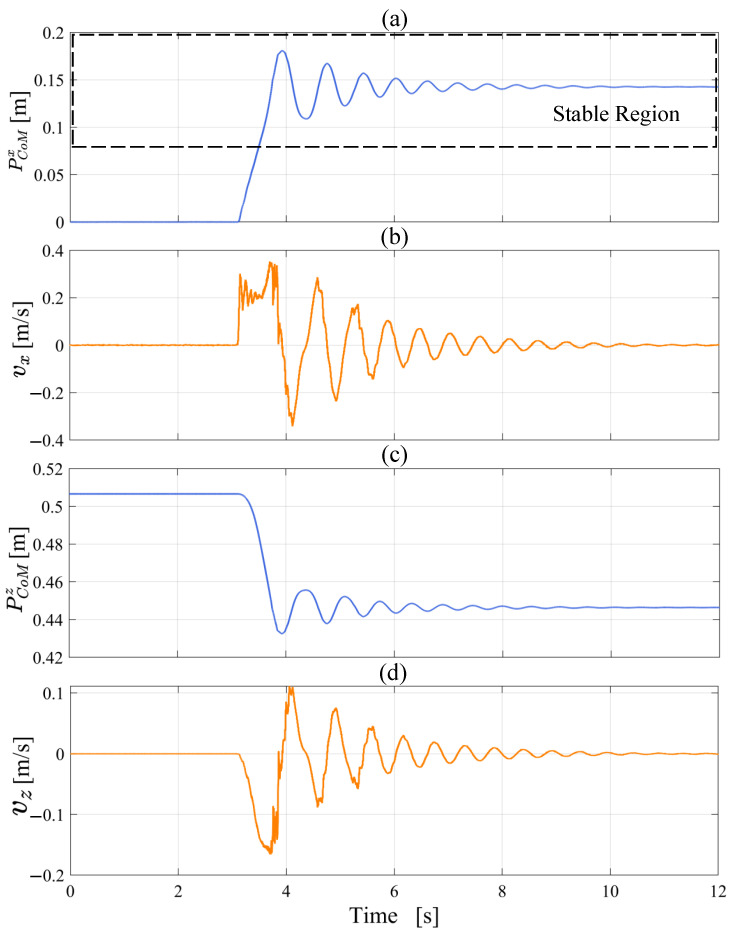
The position and velocity data of the CoM during the kneeling down experiment (**a**–**d**).

**Figure 16 micromachines-13-01252-f016:**
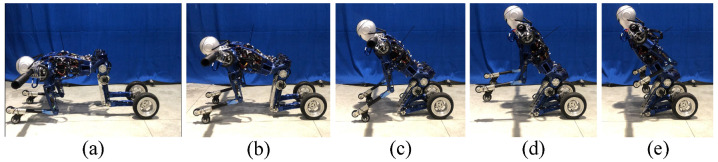
The transition of robot posture from crawling on all limbs to kneeling: (**a**) Crawling on limbs; (**b**–**e**) From crawling to kneeling.

**Figure 17 micromachines-13-01252-f017:**
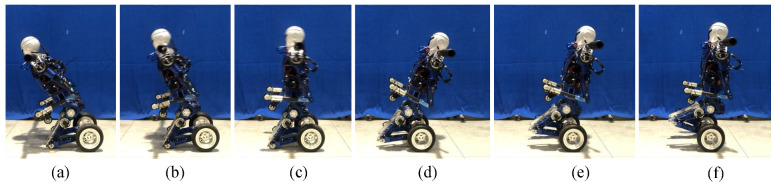
Mode transition experiment from kneeling to balanced standing (**a**–**f** ).

**Figure 18 micromachines-13-01252-f018:**
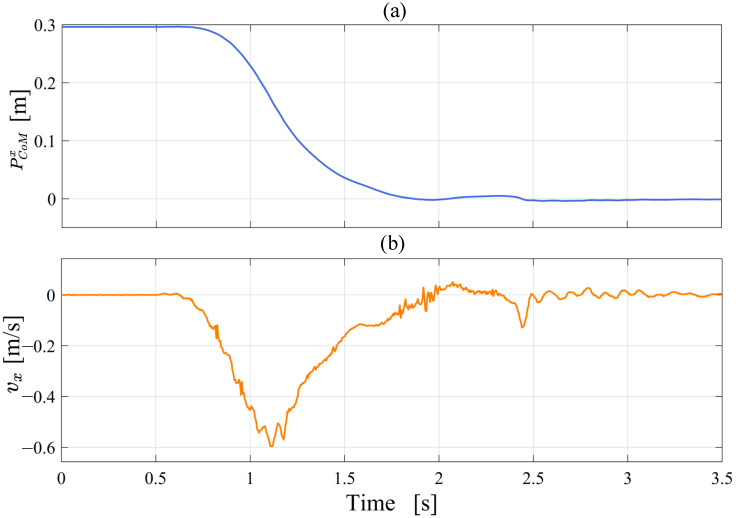
The position and velocity data of the CoM during the modal transition from kneeling to balanced standing (**a**,**b**). $P_{CoM}^{x}$ is the projection of the distance of the CoM relative to the coordinate system $\sum_{w}^{}$ on the *x*-axis and $v_{x}$ is the velocity of the CoM in the *x*-axis direction.

**Figure 19 micromachines-13-01252-f019:**
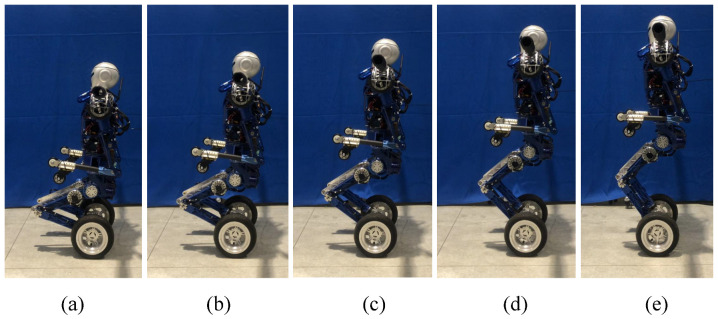
The experiment of the robot standing up (**a**–**e**).

**Figure 20 micromachines-13-01252-f020:**
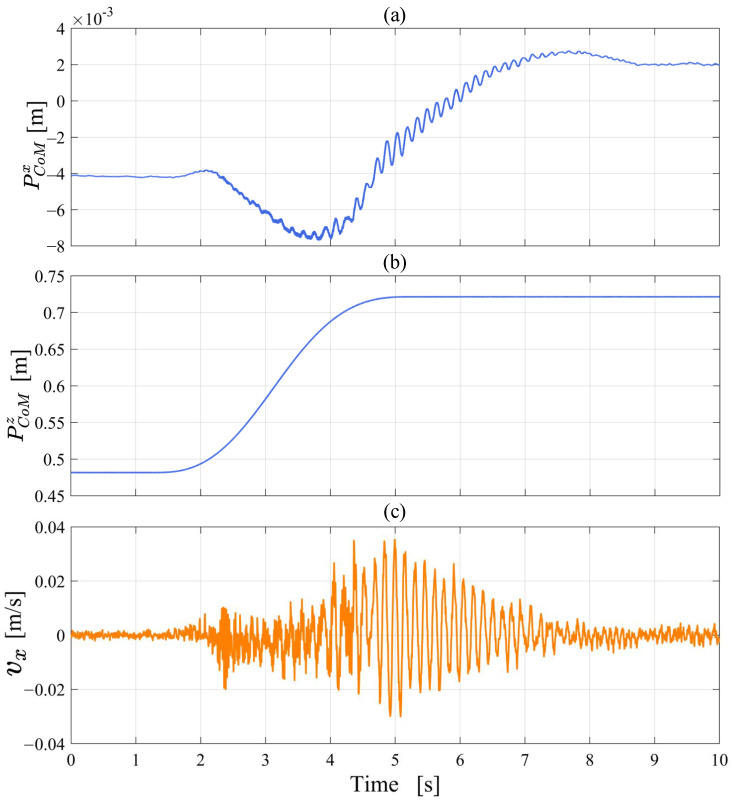
The position and velocity data of the CoM relative to the coordinate system $\sum_{w}^{}$ during standing up (**a**–**c**).

**Table 1 micromachines-13-01252-t001:** BHR-W joints angle range.

Joint	Angle Range (deg.)
Shoulder pitch	−90, +90
Shoulder roll	0, +80
Elbow pitch	+30, +150
Hip pitch	−90, +30
Hip roll	−20, +60
Knee pitch	0, +135

**Table 2 micromachines-13-01252-t002:** Parameters of the upright balanced moving model.

Symbol	Parameter Name	Value
$M_{b}$	Mass of the body	51.5 kg
$M_{w}$	Mass of the wheel	1.75 kg
$R_{w}$	Radius of the wheel	127 mm
$L_{1}^{leg}$	Length of the thigh	350 mm
$L_{2}^{leg}$	Length of the calf	350 mm
*l*	Length of the upper body	780 mm
$L_{3}$	Distance between two wheels	328 mm
$I_{w}$	Moment of inertia of wheel	0.0142 kg·m^2^
*h*	Height of CoM	-
$h_{ref}$	Reference height of CoM	-
$I_{\theta }$	Moment of inertia of body around *y*-axis	-
$I_{\gamma }$	Moment of inertia of body around *z*-axis	-
*v*	Forward velocity	-
$v_{ref}$	Reference forward velocity	-
$v_{turn}$	Velocity compensation for turning	-
${\dot{\phi }}_{L}$	Rotation speed of left wheel	-
${\dot{\phi }}_{R}$	Rotation speed of right wheel	-

**Table 3 micromachines-13-01252-t003:** Parameters of the crawling motion model.

Symbol	Parameter Name	Value
$L_{1}^{arm}$	Length of the upper arm	400 mm
$L_{2}^{arm}$	Length of the forearm	320 mm
$L_{1}^{leg}$	Length of the thigh	350 mm
$L_{2}^{leg}$	Length of the calf	350 mm
$L_{torso}$	Length of the torso	500 mm
$H_{aw}$	Height from ground to wrist	110 mm
$H_{torso}$	Height of the torso	-
$R_{turn}$	Radius of turning	-

## Data Availability

Not applicable.

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
