# Peer review of "Upright and Crawling Locomotion and Its Transition for a Wheel-Legged Robot"

_micromachines, 2022, doi:10.3390/mi13081252_

Round 1
Reviewer 1 Report
I suggest the major revision, because the paper lacks any reasoning for the general design of the robot. Why is it humanoid-like? Is this design really optimal for the proposed walking and crawling performance? For me, this is doubtful. For example, the crawling posture depicted on Fig. 5(a) looks non-optimal in that it needs to passively hold torques in the knee and the elbow, which could be easily avoided if the knee and the elbow were reverted in such a way that the thigh and the arm are horizontal, while the shank and the forearm are vertical.
Author Response
Thanks for the comments.
Point 1: I suggest the major revision, because the paper lacks any reasoning for the general design of the robot. Why is it humanoid-like? Is this design really optimal for the proposed walking and crawling performance? For me, this is doubtful.?
Response 1: The reasoning for the general design of the robot is supplemented in the introduction section.
Point 2: For example, the crawling posture depicted on Fig. 5(a) looks non-optimal in that it needs to passively hold torques in the knee and the elbow, which could be easily avoided if the knee and the elbow were reverted in such a way that the thigh and the arm are horizontal, while the shank and the forearm are vertical.
Response 2:It is true that the posture of the robot when walking and crawling is not optimal in terms of the energy efficiency of the joints, but it has certain advantages in terms of stability and control of the system. For example, in the upright balanced moving mode, the CoM of the robot is lower in the bent-leg posture relative to the straight-leg posture, which can improve the stability of the robot system. And the bent-leg posture can adjust the upper body posture by bending and extending the legs when facing uneven ground. In the crawling mode, we hope to use the passive wheels under the wrist, because they are omni-directional wheels and easier to assist the robot in turning. And in order to cope with low passages with different heights, the crawling posture shown in Figure 5(a) can more conveniently plan the movement of four limbs to change the height of the upper body from the ground.
Reviewer 2 Report
1. As line 12, what are the feasibility of the control strategies and algorithms proposed in this paper ? Please state details in main content.
2. There are many undefined symbols in figure 4. Please well define each symbol in figure 4.
3. There is no mention of optimal input variable u in figure 4. Please state the optimal input variable u in figure 4.
4. Please state what is the definition of optimal control variable u(t)*?
5. Please state what the relation between u and output torque.
6. Please state what is the P in equation (8)?
7. Please state how to obstacle avoidance in this research.
8. Please state why the robustness in your experimental results? Do you have any modeling uncertainties and environmental disturbances in your experiment?
Author Response
Thanks for the comments.
Point 1:As line 12, what are the feasibility of the control strategies and algorithms proposed in this paper ? Please state details in main content.
Response 1: "Feasibility" may lead you to misunderstand what we mean. What we want to express is that the experimental results prove that the control strategies and algorithms proposed in this paper is effective. So we replace "feasibility" with "effectiveness".
Point 2:There are many undefined symbols in figure 4. Please well define each symbol in figure 4.
Response 2: We have supplemented the notation of undefined symbols in Fig. 4.
Point 3:There is no mention of optimal input variable u in figure 4. Please state the optimal input variable u in figure 4.
Response 3: In the original Figure 4, the torque command "τ" sent from the LQR controller to the wheel motors is the optimal input variable u. In order to avoid misunderstanding and make the controller frame clearer, we have modified Figure 4 and replaced "τ" with "u".
Point 4:Please state what is the definition of optimal control variable u(t)*?
Response 4: To make the description of u(t)* more understandable, we further explain the definition of u(t)* in this paper.
"The problem solved by LQR controller is to obtain the optimal feedback gain matrix K so that there is an optimal control variable u(t)* = -Kx(t) to minimize the quadratic performance index function:"
Point 5:Please state what the relation between u and output torque.
5: In this paper, it is considered that the optimal input variable u is equal to the output torque of the wheel motor under the premise that there is no energy loss caused by the internal friction of the wheel motor.
Point 6:Please state what is the P in equation (8)?
Response 6: P is the solution of the Riccati Equation (9).
Point 7:Please state how to obstacle avoidance in this research.
Response 7: In this paper, the upright balanced moving and crawling experiments are all controlled by the operator to move and avoid obstacles through a remote computer. In the future, we plan to install a vision system on the robot to actively identify and avoid obstacles.
Point 8:Please state why the robustness in your experimental results? Do you have any modeling uncertainties and environmental disturbances in your experiment?
Response 8: Our balance controller shows robustness in experiments because it can maintain a stable balance of the robot even if the height of the robot's CoM is changing and there are errors in the model mass distribution.
The balance controller combines the upper body motion controller and the LQR controller to update the feedback gain matrix K every cycle to achieve stable balance control in response to the robot's centroid height variation in standing up experiment, which proves that the balance controller has a certain robustness. In addition, there is a certain error in the mass distribution between the CAD model and the robot prototype, and the experimental results show that the effect of controller is less disturbed by the mass distribution error. Environmental disturbances are mainly uneven outdoor roads.
And we have supplemented the explanation of the robustness of the balanced controller in the analysis of the experimental results.
Reviewer 3 Report
1. Provide manufacturer's details for The wheel-legged robot BHR-W.
2. Is this robot designed and fabricated by your team? Is it a contribution of this paper?
3. Manufacturer or the designer must have done a kinematic analysis presented in the manuscript without that, such robots can not be brought into the market. What is the novelty of the work presented in this paper?
4. LQR 325 balance controller and the body motion controller are very common.
Author Response
Thanks for the comments.
Point 1. Provide manufacturer's details for The wheel-legged robot BHR-W.
Point 2. Is this robot designed and fabricated by your team? Is it a contribution of this paper?
Response 1&2: The wheel-legged robot BHR-W is designed and fabricated by our team, which is a contribution of this paper.
Point 3. Manufacturer or the designer must have done a kinematic analysis presented in the manuscript without that, such robots can not be brought into the market. What is the novelty of the work presented in this paper?
Response 3: We supplement the kinematic analysis in this paper, including the range of movement of the limbs and the relationship between elbow angle and torque at rated torque condition.
The novelties in the mechanical structure of the wheel-legged robot are the integrated design scheme that integrates the motor and reducer into the leg support structure and the ball screw torque transmission scheme of the elbow joint.
Point 4. LQR 325 balance controller and the body motion controller are very common.
Response 4: This paper does not take the balance controller as an innovation point. Because the balance controller is part of the transition control between upright moving mode and crawling mode, LQR balance controller and body motion controller are introduced in this paper.
Round 2
Reviewer 1 Report
Dear Editor and Authors,
the manuscript became better. I suggest only minor improvements concerning the two newly added Figures:
1) The new Fig. 2 (cloud maps) requires a more detailed caption. It should explain, what is the point (0,0,0) - is it the shoulder/hip joint, around which the end-point of the arm/leg moves to form the cloud? This point (0,0,0) should be depicted in the cloud maps.
2) In the new Fig. 4, both in the caption and on the graph, the elbow joint angle (abscissa) should be labeled as Ï´.
For clarity, the vertical axis (ordinate) should represent the lever arm instead of the torque. Then, the graph as a whole is a part of sinusoid curve, isn't it?
Reviewer 2 Report
Thanks for your response.